# 3D Reconstruction Using 3D Registration-Based ToF-Stereo Fusion

**DOI:** 10.3390/s22218369

**Published:** 2022-11-01

**Authors:** Sukwoo Jung, Youn-Sung Lee, Yunju Lee, KyungTaek Lee

**Affiliations:** Contents Convergence Research Center, Korea Electronics Technology Institute, Seoul 03924, Korea

**Keywords:** 3D reconstruction, ToF camera, stereo camera, sensor fusion

## Abstract

Depth sensing is an important issue in many applications, such as Augmented Reality (AR), eXtended Reality (XR), and Metaverse. For 3D reconstruction, a depth map can be acquired by a stereo camera and a Time-of-Flight (ToF) sensor. We used both sensors complementarily to improve the accuracy of 3D information of the data. First, we applied a generalized multi-camera calibration method that uses both color and depth information. Next, depth maps of two sensors were fused by 3D registration and reprojection approach. Then, hole-filling was applied to refine the new depth map from the ToF-stereo fused data. Finally, the surface reconstruction technique was used to generate mesh data from the ToF-stereo fused pointcloud data. The proposed procedure was implemented and tested with real-world data and compared with various algorithms to validate its efficiency.

## 1. Introduction

Sensors using computer vision systems usually provide both color and depth information. To achieve accurate depth information, the fusion of multiple sensors or expensive sensors is needed. The depth information can be applied to various computer vision solutions, e.g., Virtual Reality (VR), Extended Reality (XR), Metaverse, and 3D map reconstruction.

There are many devices and algorithms for real-time depth map acquisition, including active and passive camera systems. Active sensors include structured light cameras and well-known Time-of-Flight (ToF) sensors, while passive cameras include stereo cameras. ToF cameras compute the depth by sending an electromagnetic wave signal and measuring the phase shift of the reflected signals. Even if the ToF sensor robustly estimates the 3D information and has compact configuration, it has a high level of noise and low performance on irreflective surfaces. Additionally, the ToF camera cannot extract the color information of the scene. On the other hand, the stereo camera computes the depth by finding the two corresponding pixels in the two images acquired by the two calibrated cameras. The stereo vision system only uses a simple camera setup, and it is widely used for the high-resolution-range image. However, the stereo vision system is highly affected by the texture of the scene, and it has low accuracy relative to the active sensors.

An acquisition system composed of a ToF camera and a stereo pair is proposed in some research studies [1]. Jung et al. [2] combined two sensors by using epipolar geometry to reduce the error caused by a moving object. They also used IMU to compensate for the movement of the camera module [3]. Some studies used confidence measurements for calculating reliability of the two depth data [4,5,6]. Gandhi et al. [7] combined two systems by converting the ToF depth measurement into disparity maps to use it as initial correspondences for a stereo matching algorithm. Furthermore, a neural network was used for the disparity map fusion of a ToF sensor and a stereo camera [8]. However, these algorithms only used 2D images obtained from each sensor for fusion and have limitations using 3D information.

This paper proposes a method to obtain accurate depth maps using both 3D information and 2D images acquired by a ToF sensor and a stereo camera.

## 2. Approach to The Proposed ToF-Stereo Fusion Method

Even if the 3D pointcloud registration has high computational cost, it can obtain more detailed transformation using information such as surface, normal, curvature, and sub-pixel scale vertex. As the depth image contains 3D information, we established the idea to use 3D pointcloud data for the fusion of ToF-stereo depth maps.

The depth map produced by a stereo camera has a high resolution and inaccurate 3D data compared to the ToF depth map. To reduce the error of the raw data, we fused the two data with a 3D registration procedure. As the 3D registration process uses the 3D information of the pointcloud, the alignment error becomes smaller. The overall procedure of the proposed ToF-stereo fusion method is shown in Figure 1.

The initial transformation matrix for the two 3D pointcloud data extracted from each sensor is calculated from the extrinsic camera parameters of the two sensors. Then, the 3D registration is applied for the fine tuning. The aligned 3D data are then projected to the stereo camera image plane to extract depth map from the fusion data. As the 3D data from the two sensors have different viewpoints, the depth map has a sparse data region that can be considered as a hole. We used simple hole-filling algorithm to recover the full depth map of the fusion data. After the high-resolution depth map is generated, the color image is used for the texture mapping on the depth information. Finally, the ToF-stereo fused depth map and 3D pointcloud are obtained using the proposed method.

The next sections of this paper are organized as follows. Section 3 explains the details of the proposed method, and Section 4 shows the experimental results of the proposed method. Finally, the concluding remarks are given in Section 5.

## 3. Proposed Method

### 3.1. Camera Calibration

Camera calibration is performed by capturing images of calibration object (e.g., camera calibration board) from various viewpoints. If images are taken by the same camera with fixed internal parameters, correspondences between images are used to calculate the internal and external parameters of cameras and images [9]. As the correct corresponding pixels are hard to detect, the calibration result is not always reliable [10]. In many cases, the overall performance of the depth-sensing system is affected by the accuracy of the camera calibration result. Therefore, the stereo camera and the ToF camera must be properly calibrated together in order to extract 3D information and fuse each data point acquired from the two sensors.

The ToF camera obtains an amplitude image $I_{T}$ and a depth image $D_{T}$, while the stereo camera obtains RGB images $I_{L}$ and $I_{R}$. The depth map calculated by the stereo camera is represented by $D_{S}$. Concerning the camera projection properties, the classical Heikkila model [11] is considered for all 3 cameras. The intrinsic parameters can be estimated by standard calibration algorithms. The extrinsic parameters can be calculated by the estimation of the camera pose [12]. A camera reference frame is associated with each of the 3 cameras. The world frame is considered to coincide with the reference frame of $D_{T}$. The calibration of the extrinsic parameters is the estimation of the relative rotations and translations between the ToF and stereo camera reference frames {S, T}.

Given the set of points $P_{S}^{i}$ with respect to the left RGB image reference frame, and $P_{T}^{i}$ with respect to the ToF intensity image reference frame, the estimation of the transformation matrix *M* can be calculated by minimizing the sum of Euclidean distance errors between all corresponding points as follows [13].
(1)$$arg\underset{M_{\;}}{\min}\overset{n}{\underset{i=1}{\sum }}∥P_{T}^{i}-M\cdot P_{S}^{i}∥{}_{2}$$

The calibration process is performed and tested with sensors mounted on our rigid platform, as shown in Figure 2.

### 3.2. 3D Registration

The two pointcloud data can be aligned using a 3D registration technique. There were many studies in the 3D registration field. Men and Pochiraju [14] used image-based color information for the pointcloud registration. In [15], range image and color image are used simultaneously for the matching algorithm. Kim et al. [16,17] utilize the systematic bias noise to improve the accuracy of 3D registration.

The depth data have error characteristics that consist of systematic bias and random noise. The depth measurement model of the sensor is as follows.
(2)$$p(z\;|x)\;\sim \;N\;\left(z;x+b\left(x\right),\;\sigma^{2}\left(x\right)\right)$$
where *z* is the depth measurement along a ray, *x* is the true distance along the ray, and *b*(*x*) is the systematic bias, which in practice mostly depend on the true distance.

Even if we applied calibration technique to align the two data, the non-trivial error characteristics of the sensor will result in misalignment of the data. Therefore, we run the registration algorithm to perform fine alignment to reduce the error of the data. We use ICP with RANSAC for the fine 3D registration [15].

The two pointcloud data are combined to obtain a coarse model. Then, the initial model is defined by the method of a probabilistic multi-view fusion framework. As we considered the depth measurements $z_{T}$ and $z_{S}$ from each sensor as independent, we can merge them by joint occupancy probability as follows.
(3)$$\log\frac{p(m_{x}|z_{T})}{1-p(m_{x}|z_{T})}+\log\frac{p(m_{x}|z_{S})}{1-p(m_{x}|z_{S})}\;$$
where $p(m_{x}|z_{\;})$ means a model inferring a probability of occupancy for each voxel in space, 0 represents that the voxel is completely empty, and 1 represents a fully occupied voxel.

After fitting of the two scanned data, the surface hole is filled to enhance the surface quality. The pointcloud is ready to be used for the surface reconstruction process.

### 3.3. 3D Reconstruction Method

The acquired depth map can be converted to pointcloud data. Pointcloud is used to reconstruct the 3D surface using various methods. Lee et al. [18] used as-build model to reconstruct the map of plant pipeline. Song et al. [19] used non-uniform rational B-spline surface reconstruction with neural network PointNet.

As the two pointcloud data are jointly calibrated in our procedure, the reconstruction procedure is organized by four steps:The depth map acquired from the ToF sensor is projected to the referenced stereo camera viewpoint. As the ToF captures low-resolution images, it is necessary to interpolate the depth map with the high-resolution depth map image aligned with the stereo camera lattice;The two pointclouds are extracted from the two depth maps, and they are aligned using the ICP (Iterative Closest Point) algorithm. The pointcloud from the stereo camera is set as target, and the pointcloud from the ToF camera is set as the source pointcloud;The two pointclouds are fused and reprojected to the color camera viewpoint. As the point cloud is reprojected to the different viewpoint of the ToF camera, the depth map includes a sparse hole-looking depth map, as shown in Figure 3. The hole in the depth map is filled with the simple hole-filling algorithm [15];The fused depth map of ToF and the stereo camera is colored with texture mapping of the corresponding color image.

We proposed a sensor fusion method that uses both 2D and 3D information. In the initial surface reconstruction, we merge all depth maps ($D_{T},\;D_{s}$) into a common 3D voxel grid of occupancy probabilities. After generating the merged pointcloud data, the surface is reconstructed via Poisson equation [20].

In the next section, various methods are tested to analyze the performance of 3D reconstruction procedure including the proposed method.

## 4. Experiments and Results

### 4.1. Accuracy Test by Distance

As the accuracy of the depth map depends on the distance from the scanned object [21], we established experiments to analyze accuracy by the scanning distance. We scanned an object of known geometry and estimated the error against the ground truth. We scanned a Halcon 320 × 240 mm calibration plate (MVTec, Munich, Germany) as the known object and compared the depth map of the ToF, stereo camera, and the proposed method. We used the Root Mean Squared Error (RMSE) and Mean Absolute Error (MAE) for the evaluation metrics.

The RMSE is used to compare the 3D reconstructed objects and it is calculated as follows.
(4)$$\mathrm{RMSE}=\underset{\;}{\sum }∥P_{i}^{method}-P_{j}^{gt}∥{\;}^{2}\;$$
where $P^{method}$ is the pointcloud from each method and $P^{gt}$ is the pointcloud from the ground truth. The correspondence (*i*, *j*) of each point is matched with the closest point.

The MAE is used to evaluate the accuracy of the depth map and calculated as follows.
(5)$$\mathrm{MAE}=\underset{\left(u,v\right)\in d}{\sum }∥d_{\left(u,v\right)}^{method}-d_{\;}^{gt}∥\;$$
where $d^{method}$ is the depth map from each method and $d^{gt}$ is the tested distance.

We obtained data from 0.5 m to 10 m by 0.5 m intervals as in [22]. After the scanning process, 10 depth map images are used to calculate the average MAE, and the reconstructed 3D object is used to calculate the RMSE of the tested method. The test configuration is shown in Figure 4 and Figure 5.

The depth map of the calibration plate cannot be obtained from the ToF sensor if the object distance is more than 7.5 m. The test result is shown in Figure 6.

As the ToF sensor cannot obtain the depth of the object over the 7.5 m, the proposed method only used stereo camera 3D data for the reconstruction. Therefore, the RMSE of the proposed method is exactly the same with the stereo camera over the 7.5 m. On the other hand, since the 3D registration can be performed with the background data, the depth map is generated with shifted 3D data. That is the reason why the MAE for the depth map is affected by the ToF data. A value of 1 in MAE represents an approximately 0.04 m error at a 10 m distance. Table 1 shows the average RMSE and MAE results of the experiments.

The average RMSE and MAE is calculated with the data under the 7.5 m distance. As the average RMSE is 0.042 m for the proposed ToF-stereo fusion method which is the RMSE result between 4 m and 4.5 m, we recommend using the ToF-stereo method for scanning the 0.5 m–4.0 m ranged objects. The accuracy of the proposed method outperforms the other ToF-stereo fusion method in [20] as it shows 1.294 of average MAE.

Even though the accuracy might be unstable, as the depth map range of the stereo camera covers up to 40 m, the proposed method can acquire depth data over 10 m, which is impossible with the ToF sensor alone.

### 4.2. Accuracy Test by Full Reconstruction

We used a RTC360 (Leica, Wetzlar, Germany) laser scanning device to obtain the ground truth of the real world indoor and outdoor scene data. We also used Artec Leo (Artec3D, Niederanven, Luxembourg) to obtain the ground truth data of the small object. The tested data are shown in Figure 7.

We analyzed various methods to validate the proposed algorithm. We used ZED-mini to test the stereo camera reconstruction performance, and Azure Kinect DK (Microsoft, Redmond, WA, USA) to test the ToF reconstruction performance. For the proposed method, we used a commercial sensor ZED-mini (StereoLabs, San Francisco, CA, USA) and a ToF sensor MDC100SF (Meere Company, Seoul, Korea). The configuration of the experiment is shown in Figure 2. After collecting 10 sets of data from each sensor, the data were transferred to a personal computer and the reconstruction process is implemented with an i7-4790K processor with 16 GB memory and a Windows 10 operating system.

The performance of the algorithms is measured by the RMSE, which shows the difference from the ground truth. The RMSE is calculated with the commercial software Geomagic Wrap (Artec 3D, Luxembourg) as in Figure 8, Figure 9 and Figure 10. The green regions of the error analysis represent the ground truth value, the red color means the difference is in the (+) direction, and the blue color means the difference is in the (−) direction.

The tested indoor scene is hard to reconstruct since there is no pattern on the white wall. The average RMSE with the ToF sensor is 0.0458 m and the standard deviation is 0.0053. The average RMSE with the stereo camera is 0.1447 m and the standard deviation is 0.0231. The average RMSE with the proposed method is 0.0702 and the standard deviation is 0.0177.

The tested outdoor scene is also hard to reconstruct since there is no pattern on the floor. Additionally, the range and accuracy of the ToF sensor is reduced by the sunlight. The average RMSE with the ToF sensor is 0.0399 m and the standard deviation is 0.0250. The average RMSE with the stereo camera is 0.0147 m and the standard deviation is 0.0042. The average RMSE with the proposed method is 0.0063 and the standard deviation is 0.0016.

The character doll has a lot of feature points, and the stereo camera can reconstruct the detailed textures. The average RMSE with the ToF sensor is 0.1253 m and the standard deviation is 0.0222. The average RMSE with the stereo camera is 0.0366 m and the standard deviation is 0.0195. The average RMSE with the proposed method is 0.0110 and the standard deviation is 0.0051.

Table 2 presents the average RMSE of each 3D reconstruction method on each scene. The proposed method is the best in an outdoor scene with the character doll.

## 5. Conclusions

In this paper, we proposed a 3D reconstruction method using both ToF and stereo cameras. The proposed method is implemented and tested with various methods. The performance of the proposed method is the best in reconstruction of the outdoor scene with the character doll. Even if the ToF camera shows the best result in the indoor scene, the proposed method shows better performance than the stereo camera and shows robust reconstruction results.

One of the major attributes of the proposed method is that it improves the performance of the 3D reconstruction algorithm by using both ToF and stereo sensors. The weakness of the ToF sensor is compensated for by the stereo camera, and the weakness of stereo camera is compensated for by the ToF camera.

The limitation of the proposed method is the calibration error by the two cameras. As the test configuration is not packed with a rigid frame, the abrupt motion could cause the two cameras to vibrate separately. The result of the calibration error is the mismatch between the depth map and color image, and this results in errors in the feature matching process. This limitation should be considered in future research.

## Figures and Tables

**Figure 1 sensors-22-08369-f001:**
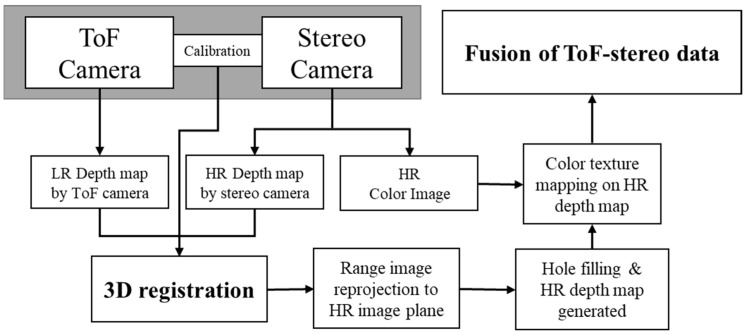
Overall procedure of the proposed method.

**Figure 2 sensors-22-08369-f002:**
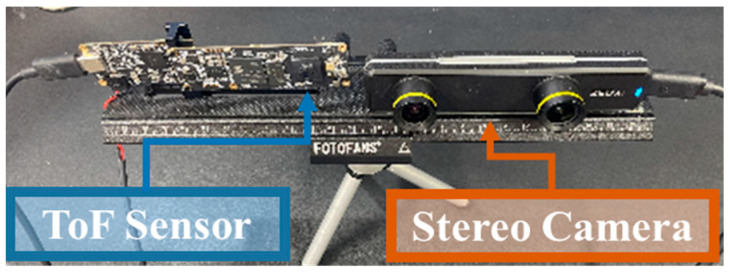
ToF-stereo fusion test configuration.

**Figure 3 sensors-22-08369-f003:**
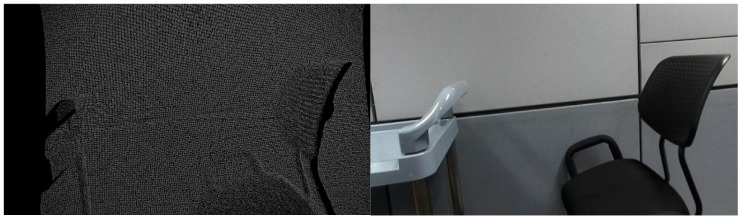
Holes in the reprojected depth map and corresponding color image.

**Figure 4 sensors-22-08369-f004:**
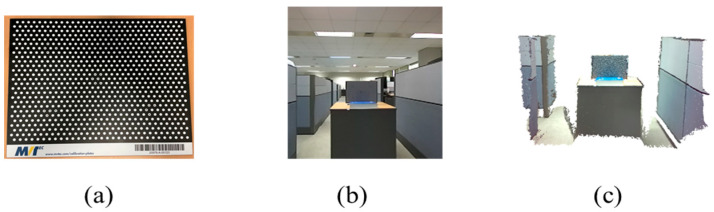
Testing accuracy by distance configuration: (**a**) Halcon calibration plate; (**b**) scanned image; (**c**) reconstructed 3D pointcloud.

**Figure 5 sensors-22-08369-f005:**
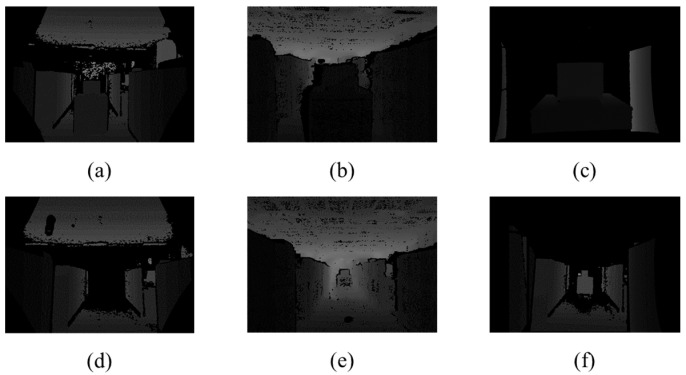
Depth map from various sensors: near distance example of (**a**) ToF sensor; (**b**) stereo camera; (**c**) proposed method; far distance example of (**d**) ToF sensor; (**e**) stereo camera; (**f**) proposed method.

**Figure 6 sensors-22-08369-f006:**
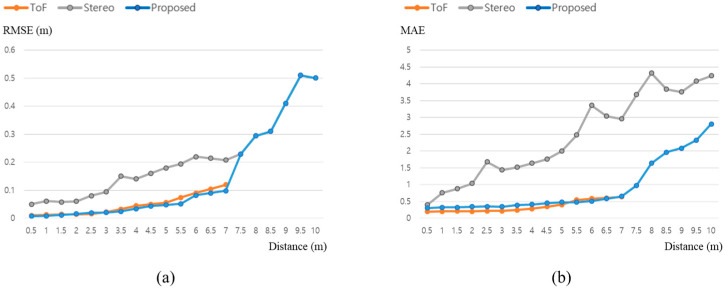
Experimental result of accuracy by distance: (**a**) RMSE; (**b**) MAE.

**Figure 7 sensors-22-08369-f007:**
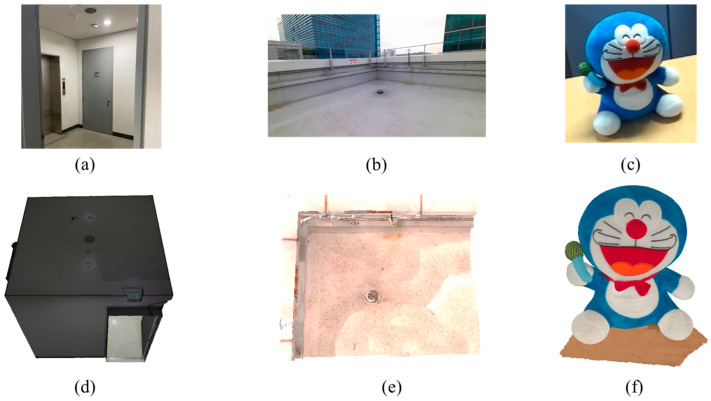
Real-world image and the reconstructed ground truth data: Image of (**a**) indoor scene; (**b**) outdoor scene; (**c**) small object (character doll); reconstructed data of (**d**) indoor scene with RTC360; (**e**) outdoor scene with RTC360; (**f**) small object with Artec Leo.

**Figure 8 sensors-22-08369-f008:**
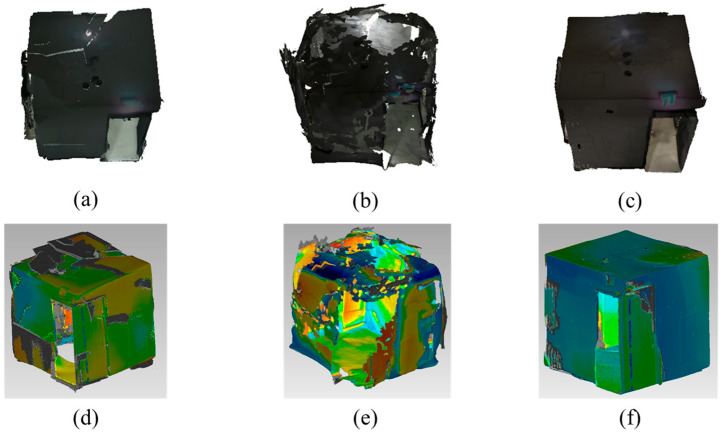
Reconstructed result and error analysis of indoor scene with various methods: Reconstructed result with (**a**) ToF camera; (**b**) stereo camera; (**c**) proposed method; error analysis of (**d**) ToF camera; (**e**) stereo camera; (**f**) proposed method.

**Figure 9 sensors-22-08369-f009:**
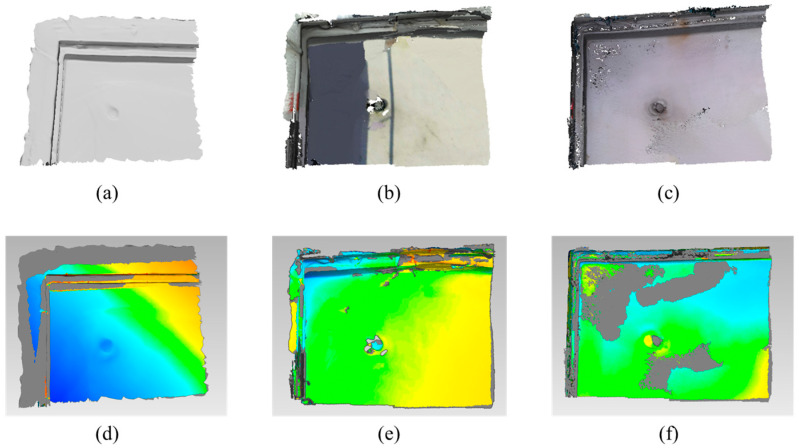
Reconstructed result and error analysis of outdoor scene with various methods: Reconstructed result with (**a**) ToF camera; (**b**) stereo camera; (**c**) proposed method; error analysis of (**d**) ToF camera; (**e**) stereo camera; (**f**) proposed method.

**Figure 10 sensors-22-08369-f010:**
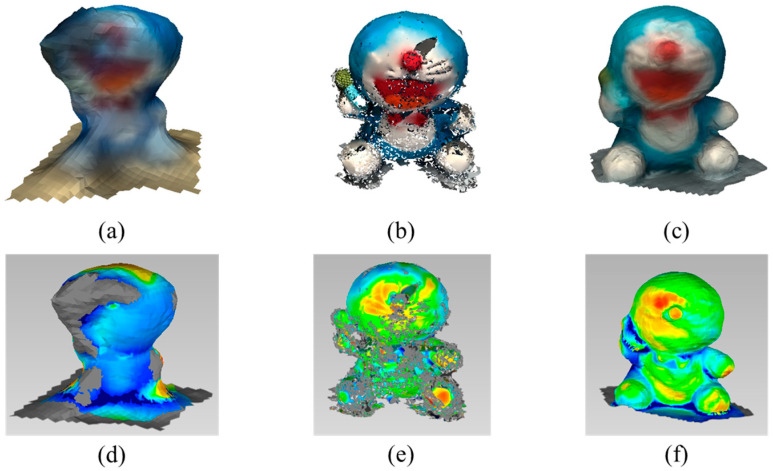
Reconstructed result and error analysis of small object with various methods: Reconstructed result with (**a**) ToF camera; (**b**) stereo camera; (**c**) proposed method; error analysis of (**d**) ToF camera; (**e**) stereo camera; (**f**) proposed method.

**Table 1 sensors-22-08369-t001:** Result of accuracy test by distance.

Method	RMSE (m)	MAE
ToF	0.047	0.355
Stereo	0.134	1.783
Proposed	0.042	0.429

**Table 2 sensors-22-08369-t002:** RMSE of reconstructed scene by various methods.

RMSE (m)	ToF Camera	Stereo Camera	Proposed Method
indoor	0.0458	0.1447	0.0702
outdoor	0.0399	0.0147	0.0063
character doll	0.1253	0.0366	0.0110

## Data Availability

Not applicable.

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
