# Peer review of "3D Reconstruction Using 3D Registration-Based ToF-Stereo Fusion"

_sensors, 2022, doi:10.3390/s22218369_

Round 1
Reviewer 1 Report
The title correctly reflects the content of the article. In this paper, the method of TOF and Stereo camera fusion is proposed, which has a complete structure and achieved good results. But there are also some problems:
1. Is the main innovation of this paper the integration of the two cameras? Which of the methods mentioned in the article is the innovation point? Is probabilistic multi view fusion framework? The description of innovation points in the article is not clear enough, and the novelty is not enough.
2. The experimental results in this paper only compared the proposed method with TOF and Stereo camera alone, but did not compare other TOF and Stereo camera fusion methods.
3. The TOF camera cannot be identified in Figure 2. It is better to mark the relevant equipment in the figure.
4. How to give and calculate the error analysis of the experimental part, it is better to have a corresponding explanation.
Author Response
Reply to the Referee # 1
“3D Reconstruction using 3D Registration based ToF-Stereo Fusion”
Sensors-1994416
First of all, we appreciate the thorough review and detailed comments made by the referee. We believe all the comments have been accommodated in the revised draft.
Comment # 1: Is the main innovation of this paper the integration of the two cameras? Which of the methods mentioned in the article is the innovation point? Is probabilistic multi view fusion framework? The description of innovation points in the article is not clear enough, and the novelty is not enough.
- We agree with the comment. We added the followings in the 3rd paragraph in page 2.
Even if the 3D point cloud registration requires high computational cost, it can obtain more detailed transformation using information such as surface, normal, curvature, and sub pixel scale vertex. As the depth image contains 3D information, we come up with an idea to use 3D point cloud data for the fusion of ToF-stereo depth maps.
- We also added the followings in the 4th paragraph in page 6.
Even though the accuracy might be unstable, as the depth map range of stereo camera covers up to 40 m, the proposed method can acquire depth data over 10 m which is impossible with the ToF sensor alone.
- We added “3D Registration based” in the title to clarify the novelty of the proposed method.
3D Reconstruction using 3D Registration based ToF-Stereo Fusion.
Comment # 2: The experimental results in this paper only compared the proposed method with TOF and Stereo camera alone, but did not compare other TOF and Stereo camera fusion methods.
- As we proposed a ToF-stereo fusion method using 3D point cloud data, we used the RMSE of the reconstructed point cloud results to show the error of the proposed method. However, the point cloud data is not appropriate to compare the depth map results by the previous other ToF-stereo fusion methods. Therefore, we added some new experiment with depth map estimation and compared the MAE and inserted the followings in the second paragraph of page 6.
Value of 1 in MAE represents approximately 0.04m error at 10m distance. Table 1 shows the average RMSE and MAE results of the experiments.
The average RMSE and MAE is calculated with the data under the 7.5 m distance. As the average RMSE is 0.042 m for the proposed ToF-stereo fusion method which is the RMSE value between the 4 m - 4.5 m, we recommend to use the ToF-stereo method for scanning the 0.5 m – 4.0 m ranged objects. The accuracy of the proposed method outperforms the other ToF-stereo fusion method in [20] as it shows 1.294 of average MAE.
Table 1. Result of accuracy test by distance
|
Method |
RMSE (m) |
MAE |
|
ToF |
0.047 |
0.355 |
|
Stereo |
0.134 |
1.783 |
|
Proposed |
0.042 |
0.429 |
Comment # 3: The TOF camera cannot be identified in Figure 2. It is better to mark the relevant equipment in the figure.
Thanks for the comment. We marked the equipment in Figure 2 with captions as follows.
Comment # 4: How to give and calculate the error analysis of the experimental part, it is better to have a corresponding explanation.
- Thanks for the comment. To make it clear, we inserted the followings in the second paragraph of page 5.
We used the Root Mean Squared Error (RMSE), and Mean Absolute Error (MAE) for the evaluation metrics.
The RMSE is used to compare the 3D reconstructed objects and it is calculated as follows.
|
|
Where is the point cloud from each method and is the point cloud from the ground truth. The correspondence (i, j) of each point is matched with the closest point.
The MAE is used to evaluate the accuracy of the depth map and calculated as follows.
Where is the depth map from each method and is the tested distance(m).

Reviewer 2 Report
The are two comments I wish to raise. The first is minor and the second is significant.
The authors did a good job of reviewing the literature. However, their review is not fully up to date and can be improved. For example, they review the works in [1] and [4] but they do not mention the most recent work of that group on the specific topic, which can be found on DOI: 10.1109/ICCVW.2017.88. The same goes for most of the reviewed works.
My most important comment regards the evaluation of results. Only deviation (RMSE) is reported but there is no report of actual error nor any study of the expression of this error depending on imaging conditions (e.g. distance). Furthermore, there is no quantitative comparison with competing works mentioned in the literature, nor a discussion on why the proposed method should be preferred over them. Hence my recommendation is to:
(a) Scan an object of known geometry and report estimation error against ground truth.
(b) Compare your results quantitatively with competing works, or at least discuss how they compare
(c) Discuss how your results vary according to imaging conditions and in particularly distance. Please also provide an optimal scanning distance for someone who wishes to use your work.
Author Response
Reply to the Referee # 2
“3D Reconstruction using 3D Registration based ToF-Stereo Fusion”
Sensors-1994416
First of all, we appreciate the thorough review and detailed comments made by the referee. We believe all the comments have been accommodated in the revised draft.
Comment # 1: The authors did a good job of reviewing the literature. However, their review is not fully up to date and can be improved. For example, they review the works in [1] and [4] but they do not mention the most recent work of that group on the specific topic, which can be found on DOI: 10.1109/ICCVW.2017.88. The same goes for most of the reviewed works.
- We agree with the comment. We added some new references including the “DOI:10.1109/ICCVW.2017.88.” as follows.
- Agresti, G.; Minto, L.; Marin, G.; Zanuttigh, P. Deep learning for confidence information in stereo and ToF data International Conference on Computer Vision Workshops (ICCVW), 2017. [CrossRef]
- Poggi, ; Agresti, G.; Tosi, F.; Zanuttigh, P.; Mattoccia, S. Confidence estimation for ToF and stereo sensors and its application to depth data fusion. IEEE Sensors Journal, 2020, 20(3), 1411-1421. [CrossRef]
- Deng, ; Xiao, J.; Zhou, S.Z. ToF and stereo data fusion using dynamic search range stereo matching. IEEE Transactions on Multimedia, 2022, 24, 2739-2751. [CrossRef]
- Tadic, ; Toth, A.; Vizvari, Z.; Klincsik, M.; Sari, Z.; Sarcevic, P.; Sarosi, J.; Biro, I. Perspectives of Realsense and ZED depth sensors for robotic vision applications. Machines, 2022, 10, 183. [CrossRef]
- Kurillo, G.; Hemingway, E.; Cheng, M.L.; Cheng, L. Evaluating the accuracy of the Azure Kinect and Kinect v2. Sensors, 2022, 22(7), 2469. [CrossRef]
- We mentioned the above 18, 19, 20 references in the 3rd paragraph of page 1 and 3rd paragraph of page 6 as follows.
Some research used confidence measurements for calculating reliability of the two depth data [4, 18, 19]. Also, neural network is used for the disparity map fusion of ToF sensor and stereo camera [20].
The accuracy of the proposed method outperforms the other ToF-stereo fusion method in [20] as it shows 1.294 of average MAE.
- We mentioned the above 21, 22 references in the first paragraph and 4th paragraph of page 5 as follows.
As the accuracy of the depth map depends on the distance from the scanned object [21], we make experiments to analyze accuracy by the scanning distance.
We obtained data from 0.5m to 10m by 0.5m interval as in [22].
Comment # 2: My most important comment regards the evaluation of results. Only deviation (RMSE) is reported but there is no report of actual error nor any study of the expression of this error depending on imaging conditions (e.g. distance). Furthermore, there is no quantitative comparison with competing works mentioned in the literature, nor a discussion on why the proposed method should be preferred over them. Hence my recommendation is to:
(a) Scan an object of known geometry and report estimation error against ground truth.
(b) Compare your results quantitatively with competing works, or at least discuss how they compare
(c) Discuss how your results vary according to imaging conditions and in particularly distance. Please also provide an optimal scanning distance for someone who wishes to use your work.
.
- Thanks for the comment. We inserted a new 4.1 section in experiment part as follows.
4.1. Accuracy test by distance
As the accuracy of the depth map depends on the distance from the scanned object [21], we make experiments to analyze accuracy by the scanning distance. We scanned an object of known geometry and estimated the error against the ground truth. We scanned Halcon 320x240mm calibration plate (MVTec, Germany) for the known object and compared the depth map of ToF, stereo camera, and the proposed method. We used the Root Mean Squared Error (RMSE), and Mean Absolute Error (MAE) for the evaluation metrics.
The RMSE is used to compare the 3D reconstructed objects and it is calculated as follows.
|
(4) |
Where is the point cloud from each method and is the point cloud from the ground truth. The correspondence (i, j) of each point is matched with the closest point.
The MAE is used to evaluate the accuracy of the depth map and calculated as follows.
|
(5) |
Where is the depth map from each method and is the tested distance.
We obtained a data from 0.5m to 10m by 0.5m interval as in [22]. After the scanning process, 10 depth map images are used to calculate the average MAE, and the reconstructed 3D object is used to calculate the RMSE of the tested method. The test configuration is shown in Figure 4-5.
Figure 4. Testing accuracy by distance configuration: (a) Halcon calibration plate; (b) scanned image; (c) reconstructed 3D point cloud.
Figure 5. Depth map from various sensors: near distance example of (a) ToF sensor; (b) stereo camera; (c) proposed method; far distance example of (d) ToF sensor; (e) stereo camera; (f) proposed method.
The depth map of the calibration plate can’t be obtained from the ToF sensor if the object distance is more than 7.5 m. The test result is shown in Figure 6.
Figure 6. Experimental result of accuracy by distance: (a) RMSE; (b) MAE.
As ToF sensor can’t obtain the depth of the object over the 7.5 m, the proposed method only used stereo camera 3D data for the reconstruction. Therefore, RMSE of the proposed method is exactly the same with the stereo camera over the 7.5 m. On the other hand, since the 3D registration can be performed with the background data, the depth map is generated with shifted 3D data. That is the reason why MAE for the depth map is affected by the ToF data. Value of 1 in MAE represents approximately 0.04m error at 10m distance. Table 1 shows the average RMSE and MAE results of the experiments.
|
Method |
RMSE (m) |
MAE |
|
ToF |
0.047 |
0.355 |
|
Stereo |
0.134 |
1.783 |
|
Proposed |
0.042 |
0.429 |
Table 1. Result of accuracy test by distance
The average value RMSE and MAE is calculated with the data under the 7.5 m distance. As the average RMSE is 0.042 m for the proposed ToF-stereo fusion method which is the RMSE result between the 4 m - 4.5 m, we recommend to use the ToF-stereo method for scanning the 0.5 m – 4.0 m ranged objects. The accuracy of the proposed method outperforms the other ToF-stereo fusion method in [20] as it shows 1.294 of average MAE.
Even though the accuracy might be unstable, as the depth map range of stereo camera covers up to 40 m, the proposed method can acquire depth data over 10 m which is impossible with the ToF sensor alone.

Round 2
Reviewer 2 Report
The authors should be commended because they did a very good job in addressing the shortcomings of the original submission, particularly in putting the work to run new experiments that quantitatively characterize the output of the proposed methods. I find that the work is improved pertinently and that the submission merits publication.